# The Extension of Vegetable Production to High Altitudes Increases the Environmental Cost and Decreases Economic Benefits in Subtropical Regions

Tao Liang [1] , Weilin Tao [1], Yan Wang [1], Na Zhou [1], Wei Hu [1], Tao Zhang [1], Dunxiu Liao [1], Xinping Chen [2] and Xiaozhong Wang [2],*

1   Chongqing Academy of Agricultural Sciences, Chongqing 401329, China
2   College of Resources and Environment, Southwest University, Chongqing 400716, China
*   Correspondence: wxz20181707@swu.edu.cn; Tel.: +86-023-68250377

**Abstract:** Global warming has driven the expansion of cultivated land to high-altitude areas. Intensive vegetable production, which is generally considered to be a high economic value and high environmental risk system, has expanded greatly in high-altitude mountainous areas of China. However, the environmental cost of vegetable production in these areas is poorly understood. In this study, pepper production at low (traditional pepper production area) and high (newly expanded area) altitudes were investigated in Shizhu, a typical pepper crop area. The output and environmental cost at the two altitudes were identified. the influence of resource inputs, climate, and soil properties on pepper production was evaluated. There were obvious differences in output and environmental cost between the two altitudes. High-altitude pepper production achieved a 16.2% lower yield, and had a higher fertilizer input, resulting in a 22.3% lower net ecosystem economic benefit (NEEB), 23.0% higher nitrogen (N) footprint and 24.0% higher carbon (C) footprint compared to low-altitude farming. There is potential for environmental mitigation with both high- and low-altitude pepper production; Compared to average farmers, high-yield farmers groups reduced their N and C footprints by 16.9–24.8% and 18.3–25.2%, respectively, with 30.6–34.1% higher yield. A large increase in yield could also be achieved by increasing the top-dress fertilizer rate and decreasing the plant density. Importantly, high-altitude pepper production was achieved despite less advanced technology and inferior conditions (e.g., a poor road system and uneven fields). It provides a reference for the study of the environmental cost of other high-altitude regions or other crop systems at high-altitude areas.

**Keywords:** life-cycle assessment; pepper; net ecosystem economic benefit; environmental cost; mitigation potential





## 1. Introduction

The global warming-driven expansion of cultivated land to high altitudes [1,2] has resulted in mountainous areas becoming an important source of vegetables and other agricultural products. To obtain a high yield and maximize economic benefits, excessive nitrogen (N) fertilizer (>350 kg ha$^{-1}$ season$^{-1}$), among other inputs, is commonly applied in high-altitude vegetable production [3]. This excessive fertilizer input imposes a significant environmental burden [4–6]. Thus, the environmental cost of vegetable production at high altitudes needs to be evaluated to facilitate sustainable vegetable production.

Substantial research has been conducted on the environmental cost of various crop production systems and cultivation patterns (e.g., open field vs. greenhouse, organic farming vs. conventional farming), among other factors (e.g., domestic vs. imported products) [7,8]. However, most of these studies were conducted in low-altitude regions (<1000 m), and the environmental cost of vegetable production at high altitudes remains unclear. There are large differences in climate, soil properties, and farm management practices between high- and low-altitude areas, which may have significant effects on

yield and environmental cost. Carbon (C) and N footprints, which are quantified as the reactive N and $CO_2$-equivalent ($CO_2$-eq) emissions per unit weight of product through life cycle assessment (LCA), can be used to assess the sustainability of agricultural systems and detect greenhouse gas (GHG) and reactive nitrogen (Nr) emission hotspots in the food production system [9,10]. However, most farmers focus predominantly on economic returns. Estimation of the net ecosystem economic benefit (NEEB) is useful for striking a balance between the economic benefits of grain yield and the environmental costs of C emissions and N loss [9,11]. High-altitude vegetable production enables residents of mountainous areas to overcome poverty by obtaining more income via the expansion of the scale of local agricultural activities. Thus, it is important to compare economic benefits and environmental costs between high- and low-altitude vegetable production areas, with consideration of the yield gap. This could lead to new methods for increasing yield and achieving environmental mitigation [12].

Increasing crop yield while simultaneously reducing the environmental cost of agriculture is one of the greatest barriers to a sustainable agricultural production system [13,14]. Large differences exist in yields among small-hold farmers due to variations in management practices. For example, a high-yield group (top 50%) obtained 59.3% more pepper than a low-yield group (bottom 50%) [15]. Other studies [16,17] have suggested that the yield gap can be significantly closed by optimizing nutrient and crop management. For example, the pepper yield was increased by 26.7% by adopting "best practice" farming methods [18]. The environmental cost differs among farmers according to crop management [19,20]. Understanding the relationship between yield and environmental cost for specific regions and crop systems is important for the development of feasible mitigation measures.

Previous studies [17,18] suggested that closing the yield gap could reduce the C footprint by 23.9–51.2%, which could be achieved by applying best-practice nutrient and crop management methods in intensive vegetable production systems. However, the environmental mitigation potential of vegetable production systems at high altitudes remains poorly understood, and the relationship between yield and environmental cost is still unclear. If vegetable production continues to extend at high altitudes, the environmental cost may increase, and NEEB will decrease in subtropical regions, then closing the yield gap could mitigate the environmental cost and raise the NEEB. Pepper (*Capsicum annuum* L.) is the most extensively grown vegetable species in China, in terms of planting area [18]. China accounts for 50.8% of pepper production worldwide (FAO, 2013). Southwest China is the major pepper production and consumption area. The objectives of this study were to: compare the differences in resource input, yield, NEEB, and N and C emission between high- and low-altitude pepper production areas; determine the potential of yield increase and environmental cost reduction of high-altitude pepper production; and identify optimal management practices.

## 2. Materials and Methods

### 2.1. Study Area and Data Sources

We conducted a survey in Shizhu County (118°04′–118°29′ E; 31°22′–32°03′ N), Chongqing Province, a typical open-field pepper production area in China. Pepper grown in this area is mainly used in condiments. Pepper production is a major industry in Shizhu County, with the planting area increasing by 21.2% during the period 2010–2017. The expansion of production to high altitudes has been an important factor in this increase. Shizhu County is a humid region with a monsoon climate. The average rainfall is 1200 mm/year. Sixty farmers were randomly selected for face-to-face interviews. The farmers were from high altitude levels (HAL, 900–1200 m; pepper growth period = April to July, monthly air temperature = 18.4 °C, average monthly rainfall = 136.9 mm) and low altitude level (LAL; 500–800 m); pepper growth period = March to June, average monthly air temperature = 22.7 °C, average monthly rainfall = 138.2 mm) areas (Table 1). We also interviewed two dealers selling seeds, fertilizers, pesticides, and plastic film.

**Table 1.** The climate during the growth stage, soil property and plant density of open-field pepper production in HAL and LAL. HAL represents the high-elevation level (500–800 m), and LAL represents the low-altitude level (900–1200 m).

| | | Unit | HAL | LAL |
|---|---|---|---|---|
| Climate during growth stage | Monthly temperature | °C | 18.6 | 22.7 |
| | Monthly precipitation | mm | 139.5 | 144.8 |
| Soil property | pH | | $5.3 \pm 1.0$ | $6.1 \pm 1.3$ |
| | Organic matter | $g\ kg^{-1}$ | $21.2 \pm 18.6$ | $13.4 \pm 5.8$ |
| | Available N | $mg\ kg^{-1}$ | $103.9 \pm 28.4$ | $89.2 \pm 33$ |
| | Available P | $mg\ kg^{-1}$ | $11.0 \pm 9.1$ | $12.8 \pm 11.3$ |
| | Available K | $mg\ kg^{-1}$ | $104.6 \pm 43.7$ | $73.1 \pm 40.7$ |
| Cultivated land | Surface slope | Degree (°) | $13.1 \pm 4.6$ | $9.8 \pm 6.4$ |
| Plant density | | $10^3$ plant ha$^{-1}$ | $41.5 \pm 8.4$ | $43.9 \pm 8.9$ |

The value represented the mean ± standard deviation (SD).

The data collected from the interviews included the transplanting date, planting area, pepper yield, planting density, fertilization date, and growth period, and the application rate of resource inputs (including fertilizers, pesticides, plastic film, and the diesel consumed by agricultural machines). Soil property data were obtained from the Soil Testing and Fertilizer Recommendation Program, which was implemented in the study region during the period 2010–2016; 106 samples were obtained from high altitudes and 170 from low altitudes.

*2.2. System Boundary*

The Nr and GHG emissions for open-field pepper production were analyzed through an LCA (from cradle to grave). The study focused on vegetable production and its system boundaries; the agricultural materials stage (MS) and arable farming stage (FS) were distinguished. The MS included the production and transport of agricultural materials applied during pepper production (inorganic fertilizer, organic fertilizer, pesticides, plastic film, and diesel consumed by agricultural machines). The impact of seeds was not considered because only small quantities were used. The FS involved the application of each agricultural input (inorganic fertilizer, organic fertilizer, pesticide, and diesel). To determine the environmental and economic impacts, we expressed the results in the following units: per ha, per ton of fresh peppers produced, and in terms of NEEB (i.e., CNY Yuan).

*2.3. Reactive Nitrogen Emissions*

The Nr emissions (kg N ha$^{-1}$) were calculated using the following equation:

$$Nr\ emission = \Sigma_{i=1}^{m} MS_{i_{Nr}} + \Sigma_{j=1}^{n} FS_{j_{Nr}}$$

where $MS_{i\text{Nr}}$ represents Nr emissions during the MS, including from the production and transportation of fertilizer, pesticides, plastic film, and diesel consumed by agricultural machines (calculated by multiplying their application rate by relevant emission factors). We used the most local and recent indicators for each emission factor. The relative quantities of pollutants emitted from fertilizers, pesticides, diesel, and plastic at the MS are listed in Table S1. The term $FS_{j\text{Nr}}$ represents Nr emission from inorganic and organic N fertilizer following their application to a vegetable field (FS), including nitrous oxide ($N_2O$) emissions, ammonia ($NH_3$) volatilization, and N leaching and runoff. The details of the emission factors of the various Nr species are given in Table S2.

*2.4. Greenhouse Gas Emissions*

The GHG emissions (kg N ha$^{-1}$) were determined by the following equation:

$$GHG\ emission = \Sigma_{i=1}^{m} MS_{i_{CO_2}} + \Sigma_{j=1}^{n} FS_{j_{CO_2}}$$

where MS$_i$CO$_2$ represents the emissions during the MS, including the production and transport of agricultural materials (inorganic fertilizer, organic fertilizer, pesticides, plastic film, and diesel consumed by agricultural machines). The term FS$j$CO$_2$ represents the GHG emissions from the FS for the inputs (inorganic fertilizer, organic fertilizer, pesticides, plastic film, and diesel consumed by agricultural machines). We applied an approach from the literature based on crops grown in the study region to quantify the GHG emissions; the details are given in Table S1.

*2.5. Environmental Damage Cost (EDC) and NEEB*

The EDC (CNY ha$^{-1}$) represents the cost of C and Nr loss, including the estimated cost of soil acidification due to NH$_3$ release and eutrophication by nitrate (NO$_3$) leaching and runoff [21,22]. It was assessed using the following equation:

$$EDC = \sum_{i=1}^{m} Nr_i \times P_{Nr_i} + GHG\ emission \times P_{CO_2}$$

where Nr$_i$ (kg N) represents the rate of *i*-th Nr loss; PNr$_i$ (CNY kg$^{-1}$ N) represents the individual cost per unit of Nr; and GHG emission (t CO$_2$-eq ha$^{-1}$) and PCO$_2$ (CNY t (CO$_2$-eq)$^{-1}$) denote the total GHG emissions and cost per unit of CO$_2$, respectively (Table S4).

Net revenue (10$^3$ CNY) was calculated as the yield revenue minus the cost of agricultural material on a per ha basis, and was calculated using the following equation:

$$Net\ revenue = Yield\ revenue - \sum_{i=1}^{m} rate_i \times P_i$$

Yield revenue was calculated by multiplying the pepper yield by the local market price of pepper. When determining the cost per ha of planted area of purchasing agricultural materials and hiring labor, *i* represents the material inputs, including fertilizers, pesticides, diesel, seeds, and labor; rate$_i$ represents the application rate of the *i*-th input category; and P$_i$ represents the purchase price of the *i*-th input category (Table S3).

The NEEB (CNY ha$^{-1}$) was calculated as follows:

$$NEEB = Net\ revenue - EDC$$

*2.6. Environmental Cost of Products and Economic Benefit Assessment*

The N and C footprints were used to express the GHG and Nr emissions per ton of pepper production, and were calculated as follows:

$$N\ footprint = Nr\ emission / Yield$$

$$C\ footprint = GHG\ emission / Yield$$

The NEEB for Nr and C emissions associated with pepper production was calculated using the following equations:

$$Nr_{NEEB} = Nr\ emission / NEEB$$

$$GHG_{NEEB} = GHG\ emission / NEEB$$

where Nr$_{NEEB}$ and GHG$_{NEEB}$ represent the CNY Yuan for Nr and GHG emissions, respectively. The Nr and GHG emissions were calculated on a per ha basis.

*2.7. Yield Gap and Environmental Cost Analysis*

The yield gap and environmental cost were analyzed via an LCA assessment and the quartering method [23], which categorized the 60 farmers into quartiles from high (quartile 1) to low (quartile 4) yield. The resource inputs and fresh yield of peppers in the four quartile groups are shown in Table 2.

**Table 2.** The resource inputs and output for pepper cultivation in HAL and LAL. HAL represents the high-altitude level (500–800 m), and LAL represents the low-altitude level (900–1200 m).

| Inventory | HAL | | | LAL | | |
|---|---|---|---|---|---|---|
| | Mean | Range | SD | Mean | Range | SD |
| Input | | | | | | |
| Total fertilizer (kg·ha$^{-1}$) | | | | | | |
| N | 297.3 | 169–484 | 73 | 289.3 | 163–539 | 76 |
| P$_2$O$_5$ | 290.7 | 72–860 | 180 | 237.9 | 83–792 | 141 |
| K$_2$O | 154.7 | 0–511 | 105 | 236.9 | 0–477 | 94 |
| Organic fertilizer (kg·ha$^{-1}$) | | | | | | |
| Organic C | 25.5 | 0–121 | 33 | 36.6 | 0–150 | 28 |
| N | 14.7 | 0–113 | 22 | 22.8 | 0–113 | 19 |
| P$_2$O$_5$ | 19.0 | 0–113 | 28 | 29.0 | 0–113 | 22 |
| K$_2$O | | | | | | |
| Inorganic fertilizers (kg·ha$^{-1}$) | 271.8 | 168–484 | 73 | 252.7 | 116–438 | 76 |
| N | 276.0 | 72–792 | 173 | 215.0 | 0–660 | 126 |
| P$_2$O$_5$ | 135.7 | 0–477 | 100 | 208.0 | 0–432 | 85 |
| K$_2$O | 1.3 | 0–2.1 | 0.5 | 1.5 | 0–3 | 0.7 |
| Pesticide (kg·ha$^{-1}$) | 52.1 | 0–120 | 13 | 53.4 | 26–109 | 13 |
| Plastic cover (kg·ha$^{-1}$) | 2.9 | 0–105 | 16 | 8.6 | 0–45 | 12 |
| Diesel (kg·ha$^{-1}$) | 0.75 | - | - | 0.75 | - | - |
| Seed | 4.5 | - | - | 4.5 | - | - |
| Cultivation labor | | | | | | |
| Output | | | | | | |
| Fresh yield (t ha$^{-1}$) | 11.1 | 7.5–17.5 | 2.4 | 13.3 | 6.8–22.5 | 3.4 |
| Net revenue (103 Yuan ha$^{-1}$) | 36.9 | 19.4–66.3 | 10.8 | 46.7 | 16.8–88.7 | 15.5 |

*2.8. Statistical Analyses*

Excel software was used to calculate the yield, resource inputs, Nr emission, GHG emissions, N footprint, C footprint, EDC and NEEB. The possibility of significant differences for the dependent variables mentioned above between the treatments (i.e., altitude level and farmer group categorized for the study area) was checked by means of an analysis of variance (ANOVA), using SigmaPlot (version 14.0; Systat Software, Inc., San Jose, CA, USA). Treatment means were compared using the least significant difference (LSD) at a 5% level of probability. In addition, linear regression models were performed to check whether fertilizer input and plant density have an influence on yield.

**3. Results**

*3.1. Yield and Resource Inputs*

The yield and resource inputs associated with open-field pepper production at two altitudes are presented in Table 2. The average pepper yield at the HAL was 11.1 t ha$^{-1}$ (range: 7.5–17.5 t ha$^{-1}$), which was 16.2% lower than the average yield at the LAL (13.3 t ha$^{-1}$; range: 6.8–22.5 t ha$^{-1}$). The N and P fertilizer application rates at the HAL were 297.3 kg N ha$^{-1}$ and 290.7 kg P$_2$O$_5$ ha$^{-1}$, which were 4.22% and 19.3% higher than at the LAL (N: 285.3 kg N ha$^{-1}$, P: 243.6 kg P$_2$O$_5$ ha$^{-1}$), respectively. In contrast, the K fertilizer application rate at the HAL (154.7 kg K$_2$O ha$^{-1}$) was 33.8% lower than at the LAL (233.6 kg K$_2$O ha$^{-1}$). The application rates of pesticide and diesel were 13.1% and 66.6% lower at the HAL compared to the LAL, respectively. The plastic film application rate was

similar between the altitudes. In summary, pepper production at high altitudes resulted in a lower yield, with higher N and P fertilizer applications.

### 3.2. Nr Emissions, N Footprint, and NrNEEB

When expressed on a per ha of planted area basis, the average Nr emissions were comparable at the HAL (84.4 kg N ha$^{-1}$) and LAL (80.6 kg N ha$^{-1}$) (Figure 1). However, there were significant differences in the N footprint and Nr$_{NEEB}$ between the two altitudes. The mean N footprint (7.8 kg N t$^{-1}$) at the HAL was 23.0% greater than at the LAL (6.4 kg N t$^{-1}$). The Nr$_{NEEB}$ at the HAL (2.7 kg N Yuan$^{-1}$) was 32.6% greater than at the LAL (2.0 kg N Yuan$^{-1}$). At both high and low altitudes, the nitrate-nitrogen (NO$_3$-N) emissions in the FS were the largest contributor to Nr emissions, accounting for 67% of the total. The ammonia nitrogen (NH$_3$-N) released during the FS (28%) made the second largest contribution to the total Nr emissions, while N$_2$O emissions during the FS and Nr emissions during the MS accounted for less than 5% of the total.

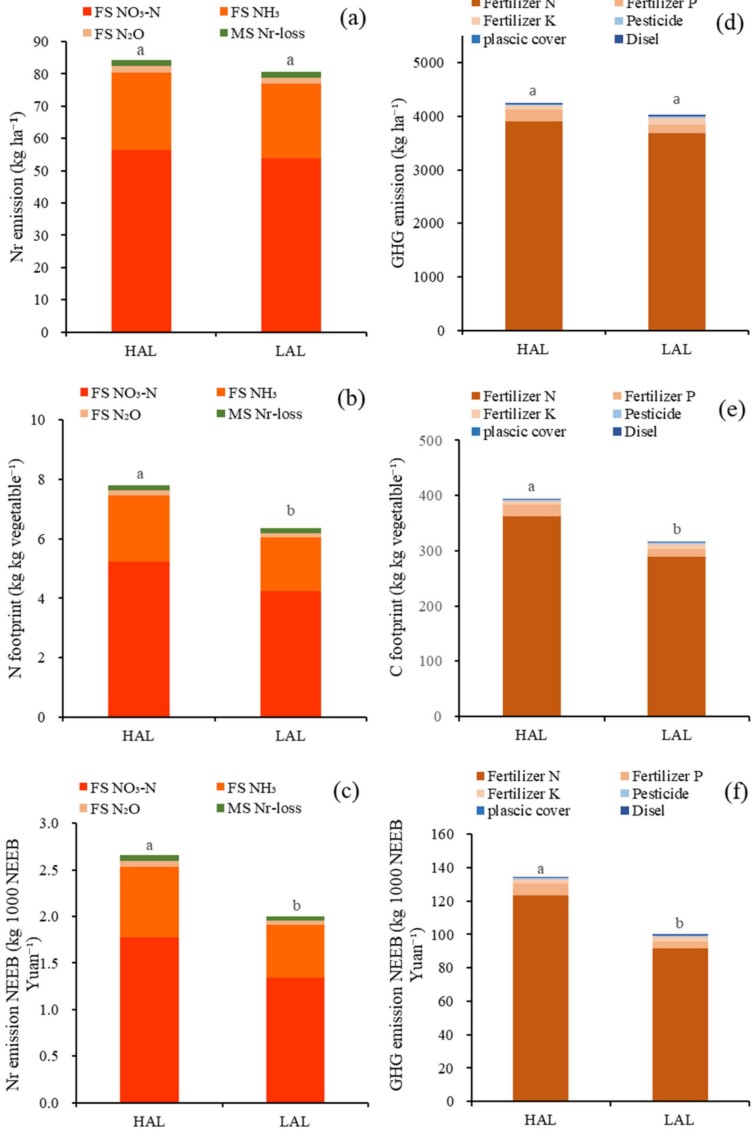

**Figure 1.** Reactive N loss (**a**–**c**) and greenhouse gas emission (**d**–**f**) on per ha, tonne of vegetable and 1000 NEEB Yuan basic of pepper production in HAL and LAL. HAL represents the high-altitude level (500–800 m), LAL represents the low-altitude level (900–1200 m). The lowercase letters represent significant differences according to least significant test (*p* < 0.05).

### 3.3. GHG Emissions and C Footprint

When expressed on a per ha of planted area basis, the GHG emissions amount at the HAL was 4240 kg $CO_2$-eq ha$^{-1}$, which was 5.26% greater than at the LAL (4029 kg $CO_2$-eq ha$^{-1}$) (Figure 1). There were significant differences in GHG emissions between the two altitudes on a product (C footprint) and NEEB (GHG$_{NEEB}$) basis; the C footprint at the HAL (393.5 kg $CO_2$-eq t$^{-1}$) was 24.0% greater than at the LAL (317.3 kg $CO_2$-eq t$^{-1}$) and the GHG$_{NEEB}$ at the HAL (134.1 kg $CO_2$-eq 10$^3$ yuan$^{-1}$) was 33.8% greater than at the LAL (100.3 kg $CO_2$-eq 10$^3$ yuan$^{-1}$) (Figure 1). The resource input that made the largest contribution to total GHG emissions was N fertilizer (91.3–92.1%); the other inputs only contributed to 7.9–8.7% of the total GHG emissions.

### 3.4. NEEB, Nr-$_{NEEB}$, and GHG-$_{NEEB}$

The EDC of pepper production is shown in Figure 2. The total EDC values at the HAL and LAL were $2.19 \times 10^3$ and $2.09 \times 10^3$ Yuan ha$^{-1}$, accounting for 7.0% and 5.6% of the total revenue, respectively. No significant difference in the EDC was observed between the two altitudes. Ammonia was the dominant contributor to the EDC, accounting for around 41% of the total; GHG emissions (34%) made the second most important contribution, followed by $NO_3$-N (24%). The other contributors ($N_2O$-direct and MS-Nr) accounted for less than 1% of the total. Due to the lower yield, the revenue at the HAL ($51.2 \times 10^3$ Yuan ha$^{-1}$) was 16.2% lower than at the LAL ($61.1 \times 10^3$ yuan ha$^{-1}$). On the other hand, agricultural material costs and the EDC were similar between the two altitudes. Thus, the NEEB at the HAL ($34.7 \times 10^3$ NEEB Yuan ha$^{-1}$) was significantly lower (by 22.3%) than at the LAL ($44.7 \times 10^3$ NEEB Yuan ha$^{-1}$). The difference between the Nr$_{NEEB}$ and GHG$_{NEEB}$ was enhanced by the large difference in NEEB between the high and low altitudes. The Nr$_{NEEB}$ and GHG$_{NEEB}$ at the HAL were 32.6% and 33.8% lower than at the LAL, respectively.

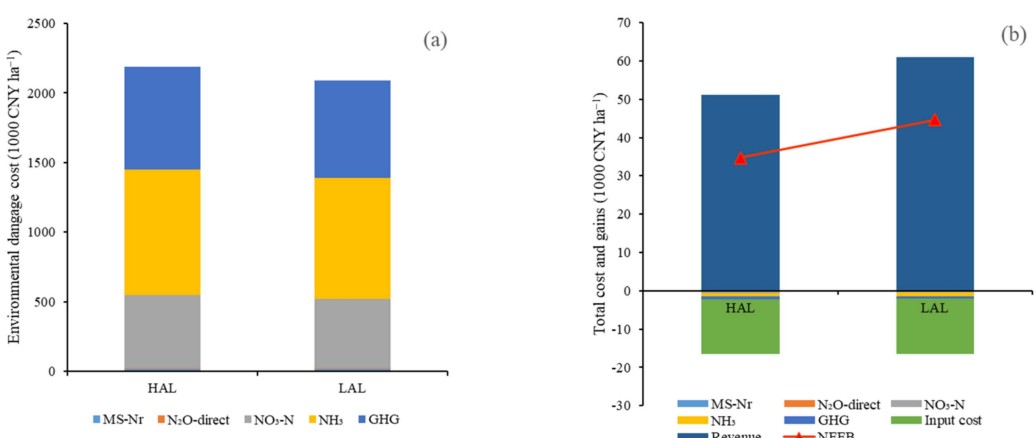

**Figure 2.** Environmental damage costs (**a**) and total net ecosystem economic benefits (**b**) in HAL and LAL for open-field vegetable production. HAL represents the high-altitude level (500–800 m), and LAL represents the low-altitude level (900–1200 m).

### 3.5. Yield Gap and Environmental Cost

There was a large yield gap among the four groups at both altitudes. Relative to quartile group 1, at the same altitude, the yields in groups 2–4 were 42.4%, 30.1%, and 21.1% lower at the HAL, and 47.8%, 35.8%, and 18.4% lower at the LAL, respectively (Table 3). The corresponding total N fertilizer application rate in group 1 (337.6 kg ha$^{-1}$) was significantly higher than in the other three groups (279.6–290.9 kg ha$^{-1}$) at the HAL (Table 3). However, the total N fertilizer (301.1 kg N ha$^{-1}$) application rate in group 1 at the LAL was comparable to that of groups 2 (306.6 kg N ha$^{-1}$) and 3 (298.8 kg N ha$^{-1}$) (Table 3). At the LAL, group 1 achieved a higher yield without any increase in the N fertilizer input. Group 1 had the greatest NEEB among the four groups at the HAL; it was 36.6, 30.2, and

$22.6 \times 10^3$ Yuan higher than for groups 2–4, respectively (i.e., 206.0%, 94.3%, and 45.4% higher, respectively). Similarly, the difference in NEEB between group 1 and the other three groups was in the range of 14.8–37.5 $10^3$ Yuan at the LAL, i.e., the NEEB for group 1 was 29.5–137.6% greater than in the other three groups (Table 3).

**Table 3.** Resource input rate and output of open-field pepper production system among different farmers groups in HAL and LAL.

| | HAL | | | | LAL | | | |
|---|---|---|---|---|---|---|---|---|
| | 4th | 3rd | 2nd | 1st | 4th | 3rd | 2nd | 1st |
| Total fertilizer | | | | | | | | |
| N (kg ha$^{-1}$) | 279.6 + 59.8 | 290.9 + 57.4 | 281.1 + 69.7 | 337.6 + 90.1 | 234.6 + 47.2 | 298.8 + 66 | 306.6 + 80.4 | 301.1 + 88.3 |
| P$_2$O$_5$ (kg ha$^{-1}$) | 278.4 + 196.1 | 293.8 + 164.7 | 312.5 + 205.5 | 278 + 165.1 | 181.4 + 52.6 | 217.1 + 180 | 290.5 + 154.2 | 285.6 + 126.4 |
| K$_2$O (kg ha$^{-1}$) | 137.9 + 94.1 | 170.8 + 92 | 117.6 + 94.9 | 192.5 + 128.2 | 223.8 + 45.9 | 203.2 + 97.2 | 214 + 116.4 | 293.3 + 82.3 |
| Inorganic fertilizer (kg ha$^{-1}$) | | | | | | | | |
| N | 267.3 + 57.6 | 261.6 + 57.8 | 260 + 74.7 | 298.4 + 94.8 | 208.4 + 48.8 | 268.6 + 76.8 | 277.1 + 84.8 | 256.8 + 75.6 |
| P$_2$O$_5$ | 271.6 + 192.5 | 276.3 + 163.3 | 295.3 + 191.8 | 261 + 159.6 | 165 + 53.8 | 167.7 + 111.3 | 273.5 + 162.6 | 253.9 + 119.6 |
| K$_2$O | 129.5 + 85 | 147.4 + 88.4 | 98.7 + 98.4 | 167.2 + 120.1 | 202.6 + 46.4 | 185.7 + 86.5 | 192.8 + 112.2 | 250.8 + 73.1 |
| Organic fertilizer (kg ha$^{-1}$) | | | | | | | | |
| Organic C | 161 + 309 | 309 + 337 | 318 + 513 | 629 + 510 | 296 + 220 | 593 + 597 | 839 + 669 | 1004 + 486 |
| N | 12.4 + 24.6 | 29.3 + 33.4 | 21.1 + 34 | 39.2 + 35.1 | 26.2 + 20.5 | 35.5 + 35.6 | 33.5 + 26.8 | 51 + 23.9 |
| P$_2$O$_5$ | 6.9 + 14 | 17.6 + 23.1 | 17.3 + 33.1 | 17 + 13.2 | 16.4 + 9.5 | 26.2 + 26.9 | 17.1 + 16.2 | 31.7 + 16.5 |
| K$_2$O | 8.5 + 16.2 | 23.4 + 30.3 | 18.9 + 35.1 | 25.3 + 25.9 | 21.9 + 15.9 | 31 + 25.6 | 21.3 + 19.7 | 41.8 + 19.9 |
| Pesticide (kg ha$^{-1}$) | 1.2 + 0.6 | 1.4 + 0.5 | 1.3 + 0.5 | 1.4 + 0.3 | 1.8 + 0.7 | 1.6 + 0.8 | 1.4 + 0.7 | 1.3 + 0.5 |
| Plastic cover (kg ha$^{-1}$) | 48 + 13.8 | 51.5 + 2.6 | 55 + 14.4 | 54 + 15.8 | 56.8 + 14.8 | 51 + 8.8 | 52.9 + 17.5 | 53 + 8.1 |
| Diesel (kg ha$^{-1}$) | 1.5 + 5.8 | 0 + 0 | 2 + 7.7 | 8 + 31 | 4.3 + 9.5 | 7.4 + 11.8 | 12.4 + 14.5 | 10.5 + 12.9 |
| Fresh yield (kg ha$^{-1}$) | 8.37 + 0.68 | 10.15 + 0.5 | 11.46 + 0.45 | 14.53 + 1.32 | 9.33 + 1.43 | 11.44 + 0.34 | 14.53 + 0.73 | 17.8 + 1.74 |
| Net revenue (1000 ha$^{-1}$) | 9.76 + 2.85 | 14.22 + 2.07 | 18.23 + 2.07 | 25.94 + 3.95 | 12.34 + 4.12 | 18.2 + 1.79 | 26.32 + 2.59 | 35.15 + 5.17 |

The value represented the mean ± standard deviation (SD).

When expressed on an area basis, at the HAL, the Nr and GHG emissions for group 1 were 15.2–15.9% and 12.5–13.9% higher than in groups 2 and 3, respectively. However, the Nr (84.5 kg N ha$^{-1}$) and GHG (4201 kg CO$_2$-eq t$^{-1}$) emissions in group 1 at the LAL were comparable to those of groups 2 (87.2 kg N ha$^{-1}$, 4403 kg CO$_2$-eq t$^{-1}$) and 3 (84.9 kg N ha$^{-1}$, 4188 kg CO$_2$-eq t$^{-1}$). When expressed on a product basis, the N "footprint gaps" at the HAL were 3.1, 1.6, and 0.5 kg N kg$^{-1}$ for groups 4, 3 and 2, respectively, i.e., group 1 contributed 48.5%, 24.9% and 7.9% less to the N footprint than groups 4, 3 and 2 at the HAL, respectively. The corresponding C footprint gaps were 170.9, 84.0, and 32.5 CO$_2$-eq kg$^{-1}$ relative to group 1 at the HAL, i.e., group 1 contributed 34.7%, 20.7%, and 9.2% less to the C footprint than groups 4, 3 and 2 at the HAL. Similarly, groups 4, 3 and 2 at the LAL had N footprint gaps of 2.4, 2.6, and 1.2 kg N kg$^{-1}$, and C footprint gaps of 124.9, 128.4, and 67.0 kg CO$_2$-eq kg$^{-1}$, compared to group 1 at the LAL, respectively. Compared to group 1, the N footprint of groups 2–4 at the LAL was 26.2%, 55.2%, and 50.5% lower, respectively, and the C footprint was 22.0%, 35.1%, and 34.5%, lower. Compared to the average values reported by the farmers surveyed at the same altitude level, the N and C footprints in group 1 were lower by 16.9% and 18.3% at the HAL, and by 24.8% and 25.2% at the LAL, respectively. The Nr$_{NEEB}$ and GHG$_{NEEB}$ gap between group 1 and the other groups was enhanced by the large NEEB gap. In group 1, the Nr$_{NEEB}$ was 15.3–92.1% lower, and the GHG$_{NEEB}$ was 17.6–98.1% lower than in the other three groups at the same altitude level.

## 4. Discussion

### 4.1. Environmental Cost of Pepper Production at Different Altitudes

Both economic and physiological factors drive the high environmental cost of vegetable production [24]. Our results demonstrated that high-altitude vegetable production resulted in higher Nr and GHG emissions than low-altitude production. On an area basis, due to the higher resource input, Nr (84.4 kg N ha$^{-1}$) and GHG (4240 kg CO$_2$-eq ha$^{-1}$) emissions were higher than reported in previous studies of other low-altitude vegetable production systems (Nr emission: 43 kg N ha$^{-1}$, GHG emission: 4638–4854 kg CO$_2$-eq ha$^{-1}$ [25]). However, the emissions were lower than reported for greenhouse vegetable production (GHG emission: 7061–19,820 kg CO$_2$-eq ha$^{-1}$ [3,26]), due to the additional structural materials (metal, plastic, irrigation facilities, etc.) and resources (electricity, water) required

in greenhouses compared to open-field production systems. Greenhouse system emissions are much higher than in grain production systems (GHG emission: 2210–3629 kg $CO_2$-eq ha$^{-1}$ [27,28]) due to the higher resource inputs. On a yield basis, the N and C footprints at the HAL were also higher than for other vegetable systems, mainly due to the lower yield. For example, the average N footprint (7.6 kg N t$^{-1}$) at the HAL was 348% higher than at the LAL (2.1 kg N t$^{-1}$) due to the higher yield at the former level (11.1 t ha$^{-1}$). This was still far below the "ox horn" pepper yield reported in the eastern plains area of China (41 t ha$^{-1}$) [29].

There was a large difference in the N and C footprints between the high and low altitudes in this study. Compared to the LAL, the N and C footprints at the HAL were 23.0% and 24.0% higher, respectively. There were several reasons for this. First, fertilizer was the main source of Nr and GHG emissions [30]. Compared to the LAL, the total N and P fertilizer input at the HAL was 4.2% and 19.3% higher, respectively, which could be attributed to the higher runoff loss associated with the steeper slope (13°, Table 1) in the arable area at the HAL [31,32]. Second, the type of fertilizer (organic or inorganic fertilizer) had an important effect on Nr and GHG emissions during vegetable production [33]. Compared to N from organic fertilizers, inorganic N fertilizer contributed 18.7–22.2% of the runoff per unit N input [34]. This resulted in the Nr and GHG emissions on an area basis being 4.6% and 5.3% greater at the HAL than the LAL, respectively. Third, yield also exerted an important influence on the Nr and C footprints [35,36], with the yield at the HAL being 16.2% lower than at the LAL. This large variation in yield was the main driver of the higher N and C footprints at the HAL. The pepper yield was higher at the LAL than the HAL. There are several explanations for this. First, the climate differs by altitude [37]. The optimal temperature range for pepper cultivation is 25–31 °C [38,39]. The temperature during the growth stage at the LAL (22.7 °C) in this study was more suitable than at the HAL (18.6 °C). Soil properties also play an important role in pepper production [28,40]. The optimum soil pH range for pepper production is 6.2–8.5 [40,41], so the pH (5.3 ± 0.96) at the HAL may have had a more adverse impact on pepper yield than the higher pH at the LAL (6.1 ± 1.3). Third, the production conditions affected the yields at the two altitudes. The HAL had a poor road system and uneven vegetable fields, both of which are very important with respect to the accessibility of yield-improving technology and agricultural machinery. For example, the poor road system at the HAL limited the application of agricultural machinery and organic fertilizer. Compared to the LAL, the diesel consumed by agricultural machinery and organic C applied at the HAL were 66.6% and 32.4% lower, respectively (Table 2).

The Nr$_{NEEB}$ and GHG$_{NEEB}$ at the LAL were 32.6% and 33.8% lower than at the HAL, mainly due to the lower yield and net revenue at the HAL. At the HAL, farmers applied excessive resource inputs to overcome the N and P runoff losses and obtain high pepper yields. Their lack of nutrient management expertise resulted in large resource input costs. The higher cost combined with lower total revenue due to the lower yield resulted in a lower net revenue at the HAL.

Overall, pepper production at high altitudes resulted in higher Nr and GHG emissions, larger N and C footprints and lower yields, net profit, and NEEB. This clearly showed that the expansion of pepper cultivation to higher altitudes was associated with greater environmental costs and relatively smaller economic benefits.

### 4.2. Potential for Mitigating Environmental Costs

The N and C footprints of group 1 were 7.9%, 24.9%, 48.5%, and 16.9%, and 9.2%, 23.7%, 48.3% and 18.3%, lower than in groups 2–4 and the average at the HAL, respectively (Figure 3). Closure of yield gaps offers great potential for mitigating the environmental cost of agricultural systems [35]. Although the Nr and GHG emissions at the HAL were the largest in group 1 in this study, mainly due to the higher fertilizer input than in the other groups, the increase in the rate of N fertilizer application was 17% lower than the rate of yield increase in group 1. This resulted in a smaller N and C footprint at the

HAL in group 1 compared to the other groups and the average for all farmers. Many studies [12,15,42] have indicated that the yield gap can be closed by improving crop and nutrient management by adopting best-practice farming methods. However, the factors responsible for the yield gap differ among regions and crops due to large variations in climate, soil conditions, and management practices. In this study, the higher yield in group 1 at the HAL and LAL could be explained by several factors. The top-dress fertilizer rate, especially the N and K fertilizer rate, was the major factor influencing vegetable yield. Our results revealed positive relationships between the N fertilizer top-dressing rate and yield. An increase in the N and K fertilizer application rates promoted rapid crop growth and fruit formation at the later growth stage [43]. There were no significant relationships between the total N, P, and K fertilizer application rates and yield (Figure 4). Second, regarding organic management, organic fertilizer application can also improve soil quality, reduce soil Nr loss, and promote crop growth [44,45]. Due to long-term and excessive inorganic fertilizer inputs, several soil problems (e.g., acidification, soil-borne disease, and decreased quantity and stability of structural aggregates) are very common in the survey area. Our results revealed a positive relationship between the organic C input rate and yield. Additional organic C inputs improve soil structure, stimulate soil microbial activity, and improve the yield of vegetable crops. Third, regarding management, an appropriate planting density will be able to fully exploit the light conditions and soil nutrients. Our results indicated that an excessive planting density would have a negative influence on pepper yield (Figure 4). This might result in undue competition between plants for limited light and heat, thereby reducing carbohydrate production [46]. In addition, there was room for further optimization of group 1. The N and P fertilizer input was much higher than the vegetable demand and recommended fertilization rate (147–200 kg N ha$^{-1}$, 75–90 kg P$_2$O$_5$ ha$^{-1}$) [47], which could be optimized with in-season root-zone nutrient management to match the nutrient supply [12]. Furthermore, the traditional N fertilizer (e.g., urea, ammonium carbonate) applied in the survey area could be substituted for an N fertilizer with a nitrification inhibitor [48,49].

In the future, it will be important to consider the environmental cost of resource inputs, and their mitigation potential in the context of vegetable production at high altitudes. For the first time, this study systemically compared the environmental cost of resource inputs for pepper production between high and low altitudes. Our results indicated that pepper production at high altitudes was more resource-intensive and led to larger N and C footprints than production at low altitudes. The results provide a reference for studies of the environmental cost of crop production in other high-altitude regions of the world. We quantified the environmental cost of pepper production at high altitudes in southwest China and identified potential mitigation measures. The results indicated that the pepper yield could be improved, while the environmental cost was greatly diminished by optimizing crop and nutrient management. These results have important implications for sustainable vegetable production in high-altitude regions. However, the uncertainty of this research is that we surveyed the Shizhu of southwest China, which is a local characteristic; for future research, we recommend wider coverage and longer-term experimentation for a higher level of generalization.

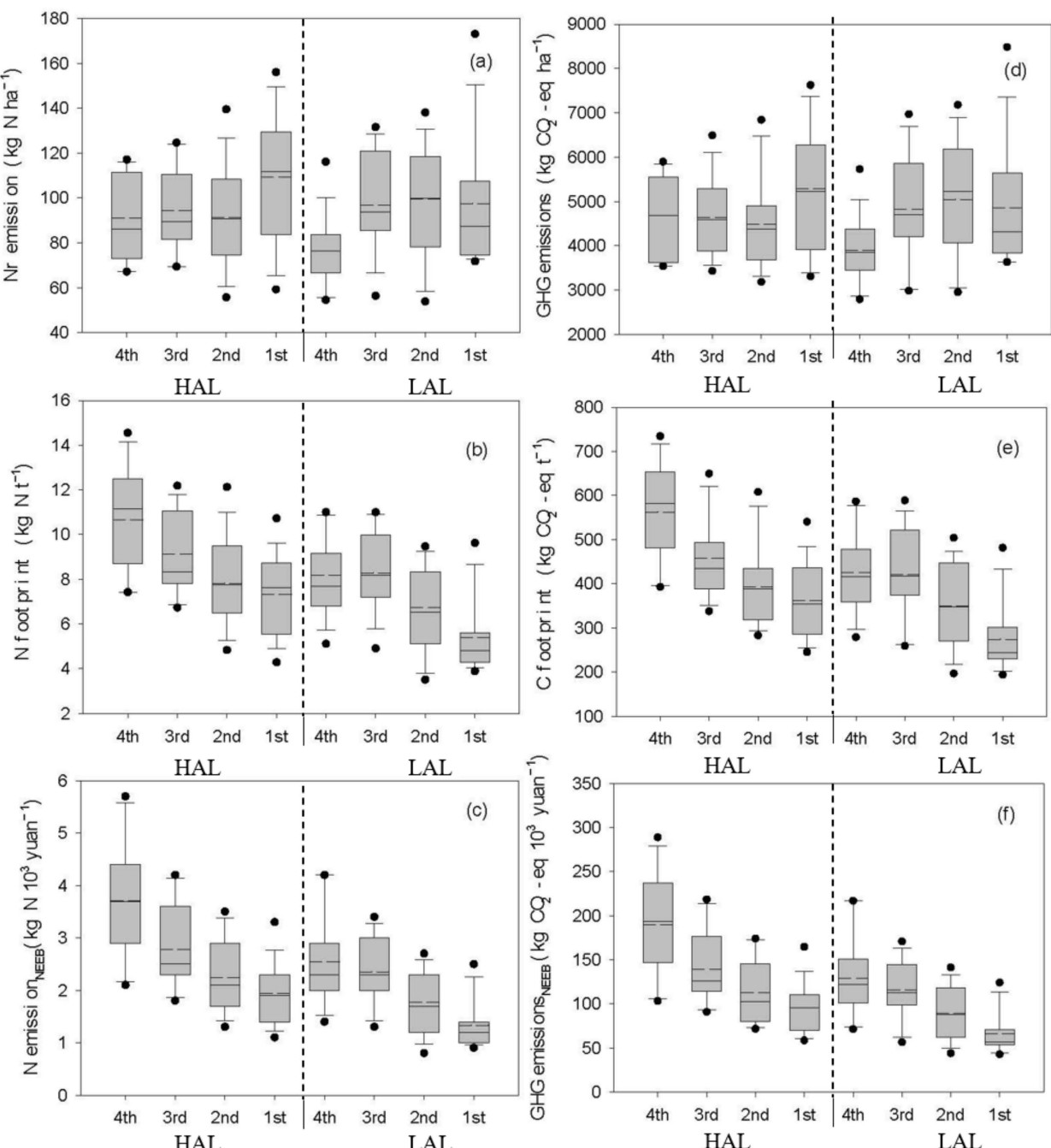

**Figure 3.** The Nr Emissions (**a**), N footprint (**b**), N emission$_{NEEB}$ (**c**), GHG emission (**d**), C footprint (**e**) and GHGemission$_{NEEB}$ (**f**) in different groups of open-field pepper production. The yield data from surveyed farmers were ordered from low to high and divided into four quartiles: 1st quartile (best 25%), 2nd quartile (50–75%), 3rd quartile (25–50%), and 4th quartile (lowest 25%). HAL represents the high-altitude level (500–600 m), and LAL represents the low-altitude level (900–1200 m).

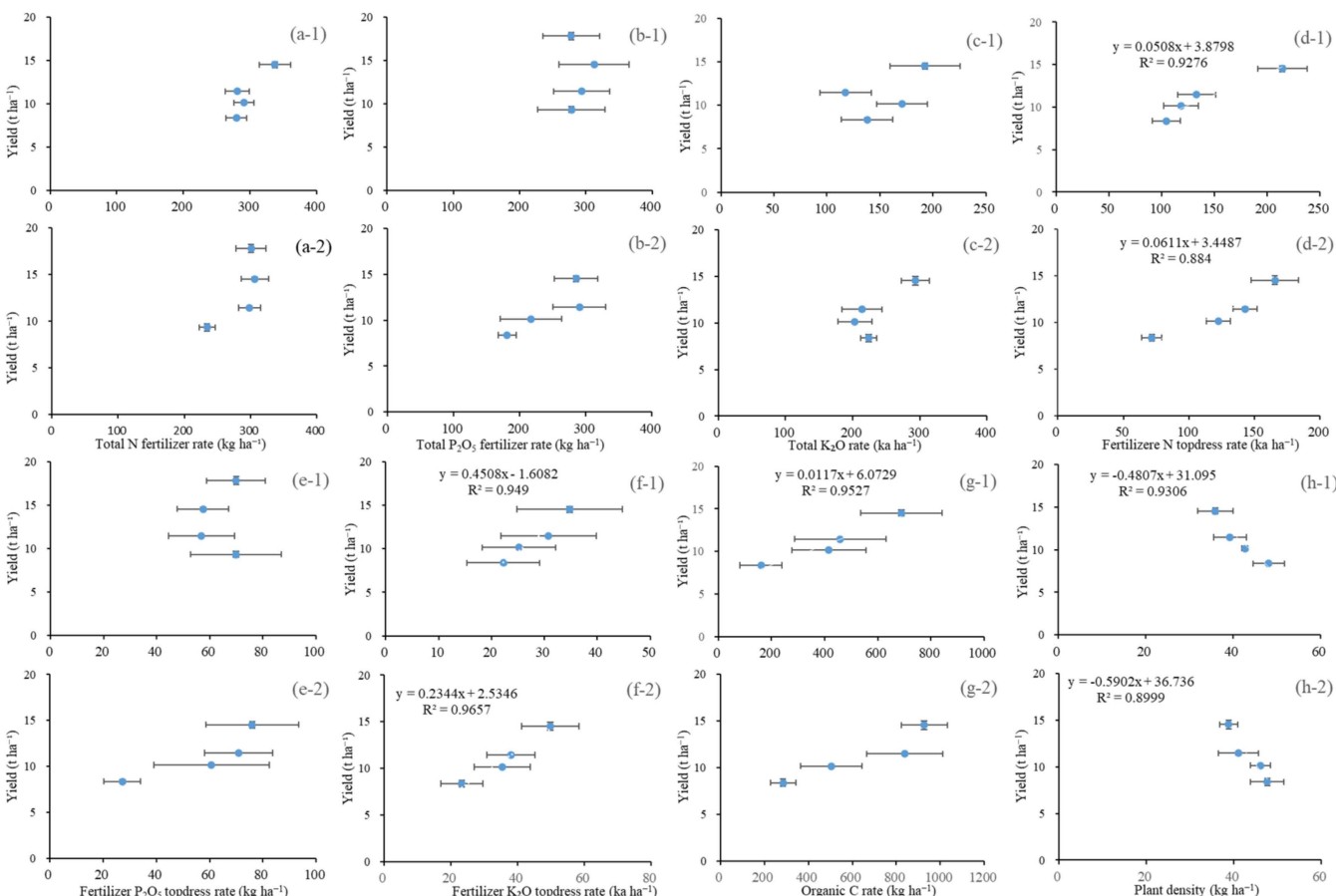

**Figure 4.** Relationships between yield and total N fertilizer rate, total $P_2O_5$ fertilizer rate, total $K_2O$ fertilizer rate, N fertilizer top-dress, $P_2O_5$ fertilizer top-dress rate, $K_2O$ fertilizer top-dress rate, Organic C, plant density in HAL (**a-1**–**h-1**) and LAL (**a-2**–**h-2**) for open-field vegetable production in southwest China.

## 5. Conclusions

The results of this study indicated that open-field pepper production at high altitudes had higher resource inputs and lower yields than production at low altitudes, resulting in 4.6% and 5.3% higher Nr and C emissions on an area basis, and 23.0% and 24.0% higher N and C footprints on a product basis. There is considerable potential for yield increases and mitigation of the environmental cost at high altitudes. Compared to the average for all farms, the high-yield group contributed 12.4% and 18.3% less to the overall C and N footprints, mainly due to their higher yields. Reducing plant density and increasing the topdressing fertilizer rate and organic C inputs could improve the pepper yield and decrease the N and C footprints.

**Supplementary Materials:** The following supporting information can be downloaded at: https://www.mdpi.com/article/10.3390/land12030662/s1, Table S1: The coefficients of reactive N (Nr) emission and greenhouse gas (GHG) emission for agricultural inputs at the agricultural materials stage (MS); Table S2: The emission factor of $NO_2$, $NH_3$, $NO_2$ and $NO_3^{-1}$ for N fertilizer inputs at the agricultural materials stage (FS); Table S3: Components EDC (CNY ha$^{-1}$) for vegetable production; Table S4: The application rate and price of various agricultural inputs for farm management under different elevation level for cabbage production. HAL represents high elevation level (900–1200 m), LAL represents low elevation level (500–800 m). The marketing price were obtained from the dealer in CNY, and converted into USD based on an average exchange rate of 7. References [50–61] are cited in the supplementary materials.

**Author Contributions:** Conceptualization, T.L. and X.W.; methodology, X.C.; Data curation, Y.W.; validation, W.H., T.Z. and N.Z.; formal analysis, D.L.; investigation, T.L.; writing—original draft preparation, T.L.; writing—review and editing, W.T.; visualization, T.Z.; supervision, Y.W.; project administration, X.W.; funding acquisition, T.L. All authors have read and agreed to the published version of the manuscript.

**Funding:** Chongqing science and Technology Bureau: CSTB2022TIAD-CUX0020.

**Data Availability Statement:** Not applicable.

**Conflicts of Interest:** The authors declare that they have no known competing financial interests or personal relationships that could have appeared to influence the work reported in this paper.

## Abbreviations

HAL represents the high-altitude level (500–800 m), LAL represents the low-altitude level (900–1200 m); LCA represents life cycle assessment; EDC represents the cost of C and Nr loss, including the estimated cost of soil acidification due to $NH_3$ release and eutrophication by nitrate ($NO_3$) leaching and runoff. NEEB represents the net ecosystem economic benefit.

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
