# Peer review of "The Extension of Vegetable Production to High Altitudes Increases the Environmental Cost and Decreases Economic Benefits in Subtropical Regions"

_land, doi:10.3390/land12030662_

Round 1

Reviewer 1 Report

Article: The extension of vegetable production to high altitudes increases

the environmental cost and decreases economic benefits in subtropical

regions

Authors: Tao Lianga, Weilin Tao, Yan Wang, Na Zhou, Wei Hu, Zhang Tao,

Dunxiu Liao, Xinping Chen and Xiaozhong Wang

The study provides relevant information regarding the environmental cost

and economic benefits of vegetable production at high altitudes.

The introduction is well designed and properly addresses to the research

topic covered in the study reflecting the current state of the art in the field

of vegetable production.

The materials and methods used to carry out the experimental research are

properly detailed.

The references cited are relevant, but a fairly large number are older than 5

years.

The results presented are consistent and discussed in a fair manner.

The results presented are consistent and discussed in a fair manner, but we

have identified some mismatches between the data contained therein and

those mentioned in the text:

- In Table 1, Table 2 and Figure 1: HAL represents the high-elevation level

(500-800 m), and LAL represents the low-altitude level (900 1200 m)?

- In Table 3: at the head of the table: HEL and LEL?

- In Figure 3: b) and c) does not contain LAL

- Figure 5 does not exist

- References are also made in the text to figures other than those discussed

and interpreted in the sentence: line199-200.

I recommend the authors to revise the names and contents of the figures

and tables so that they are consistent with the references in the text.

I have noticed that although statistical analysis is stated to be performed

there were now such referral while discussing the results and thus the claim

that the differences between the results are significant cannot be supported

statistically.

The conclusions reflect the results obtained and respond to the proposed

purpose.

The paper can in principle be accepted after revision based on the

recommended revisions.

Reviewer 2 Report

Dear Editors,

first of all, I would thanks for including me in the revision process of the manuscript submitted in Land journal n. land-2236350 and entitled: “The extension of vegetable production to high altitudes increases the environmental cost and decreases economic benefits in subtropical regions”.

The manuscript aims to investigate the potential of pepper cultivation in different areas characterized by different altitudes. Evaluation was performed based on the influence of resource inputs, climate and soil properties.

A general aspect that it should be addressed is related that the text is reported in first person, while an high quality manuscript should be write in third person with an objective way. Therefore, I suggest the Authors to check the manuscript to eliminate the terms “We” and reformulate the sentences.

INTRODUCTION

In the introduction section it is missing the hypothesis of the current study. In addition the objectives of the study are not correctly explained and need to be rewritten in order to be more clear for the potential readers of the manuscript.

MATERIALS and METHODS

The statistical analysis should be rewritten according with the experimental design adopted. Actually, it is not clear the experimental design adopted and how the treatment are managed for the statistical analysis.

Table 1 and Table 3 please specfy the meaning of the ± symbol.

In Table 3 the terms HEL and LEL are not correctly reported as in the caption is wrote HAL and LAL. In addition, if them are acronyms they meaning should be reported in the table caption.

Reviewer 3 Report

Rewiev: land-2236350

The extension of vegetable production to high altitudes increases the environmental cost and decreases economic benefits  in subtropical regions.

The article deals with an important topic. The introduction has been written correctly, well outlines the purpose of taking up the topic and contains the characteristics of the most important research previously published in the field of the presented subject. He makes some remarks to improve the manuscript:

- maybe it would be worth putting a list of abbreviations used in the manuscript at the beginning, because there are a lot of them, it would make it easier for the reader to navigate through the article

- why was this formula used for Reactive nitrogen emissions and Greenhouse gas emissions?

 - conclusions should contain not only a summary of the results obtained, but also show how the obtained results can contribute to the further development of knowledge in the field of research
